# Prevalence of Post-intensive care syndrome among intensive care unit-survivors and its association with intensive care unit length of stay: Systematic review and meta-analysis

Temesgen Ayenew[1]*, Menberu Gete[1], Mihretie Gedfew[1], Addisu Getie[1], Abebe Dilie Afenigus[1], Afework Edmealem[1], Haile Amha[1], Girma Alem[1], Bekele Getenet Tiruneh[2], Mengistu Abebe Messelu[1]

1 Department of Nursing, Debre Markos University, College of Health Sciences, 2 Department of Internal Medicine, Debre Markos University, School of Medicine

* teme31722@gmail.com; temesgen_ayenew@dmu.edu.et

## Abstract

### Background

Post-intensive Care Syndrome (PICS) is defined as various physical, psychological, and cognitive, impairments that can arise during an ICU stay, continue after leaving the ICU, or even persist following hospital discharge. It impacts both patients and their family's quality of life. Various primary studies worldwide have reported prevalence of PICS among ICU survivors. However, these studies exhibit inconsistency and wide variations. Therefore, this systematic review and meta-analysis aimed to estimate the pooled prevalence of post intensive care syndrome among intensive care unit survivors along with its association with ICU length of stay.

### Methods

We used the Preferred Reporting Items for Systematic Reviews and Meta-Analyses (PRISMA) 2020 checklist for this review. We searched PubMed/Medline, CINHAL, Embase, and Google scholar to retrieve articles. The Newcastle Ottawa Scale (NOS) was used for quality assessment of articles. The random effects model with I-squared test was used to estimate the prevalence of PICS and its association with ICU length of stay. To identify the source of heterogeneity within the included studies, meta-regression and subgroup analysis were used. We employed Egger's regression test and funnel plots for assessing publication bias. STATA version 17.0 software was used for all statistical analyses. A p-value of $< 0.05$ with 95% confidence interval was used declare statistically significant.

**Data availability statement:** All relevant data are within the paper and its Supporting Information files.

**Funding:** The author(s) received no specific funding for this work.

**Competing interests:** The authors declare that they have no competing interests.

## Results

A total of 19 articles with a population of 10179 ICU-survivors were included in this review. The pooled prevalence of PICS was found to be 54.35% (95% CI = 45.54, 63.15). In sub-group analysis by region, the highest prevalence was observed in studies done in south and north America with overall prevalence of 61.95% (95% CI = 28.33, 95.62). Among the three domains of PICS (physical, cognitive and mental domains), the highest prevalence score was observed in the physical domain with overall prevalence of 45.99% (95% CI = 34.66, 57.31). In this meta-analysis, those patients who stayed more than four days in the ICU were 1.207 [95% CI = 1.119, 1.295] times more likely to develop at least one among the three domains of PICS in the post-intensive care period than their counterparts.

## Conclusion

This systematic review and meta-analysis demonstrate a high prevalence of PICS among ICU survivors, and highlight the significant association between ICU length of stay and the development of PICS. These findings underscore the need for targeted interventions to mitigate the long-term effects of critical illness, particularly for patients with prolonged ICU stays.

## Introduction

Due to advancements in medicine and technology, the number of patients surviving after receiving critical care has increased, accompanied by a decline in intensive care unit (ICU) death rates. However, there still exist unfavorable long term outcomes for these survivors, as most ICU survivors experience long term impairments known as post intensive care syndrome (PICS) [1,2].

PICS is defined as various physical, psychological, and cognitive, impairments as well as failed social reintegration that can arise during an ICU stay, continue after leaving the ICU, or even persist following hospital discharge, affecting the long term outcomes of ICU survivors [3,4]. It is also considered when a new or worsening impairment in physical (ICU-acquired neuromuscular weakness), cognitive (thinking and judgment) or mental health status develop among survivors of critical illness [5].

PICS encompasses physical, cognitive, and mental health impairments that persist after critical illness [5]. The physical domain includes muscle weakness and functional limitations, the cognitive domain involves memory and executive function deficits, and the mental health domain encompasses anxiety, depression, and PTSD [6,7]. Given the long-term impact of PICS, understanding these domains is essential for improving patient recovery.

PICS is a highly prevalent among survivors of critical illness [8]. A proportional meta-analysis indicated that nearly 50% of ICU survivors experience cognitive impairments within one-month post-discharge, with this prevalence decreasing to around 28% at long-term follow-up, beyond one year [9]. A prospective multicenter

study found that 58% of medical ICU survivors, 64% of urgent surgical ICU survivors, and 43% of elective surgical ICU survivors developed new physical, mental, or cognitive problems one year after ICU discharge [10]. It negatively impacts patients' lives by decreasing their health related quality of life, [11] increases the burden of their caregivers and families, lowering their survival rates, and creating significant challenges for previously employed ICU survivors in returning to work [3,12–14].

PICS can significantly impact not only the survivors but also their families, potentially leading to adverse psychological outcomes like anxiety, acute stress disorder, posttraumatic stress, depression, and complicated grief. It can also hinder the family member's ability to fully engage in necessary care-giving functions after hospitalization [15].

Various factors contribute to the development of PICS among ICU survivors, including duration of mechanical ventilation, stroke, delirium, hypotension, prolonged use of sedation, hypoglycemia, female sex, under-nutrition, sepsis, being older, diminished level of consciousness, presence of polyneuropathy at ICU, and days of hospital stay after ICU discharge [14,16,17].

Previous research has reported that the prevalence rates for PICS often ranges between 25% and 70%, depending on the population studied and methodological differences [5,6]. This systematic review and meta-analysis aimed to estimate the pooled prevalence of post intensive care syndrome among intensive care unit survivors along with identifying associated factors.

The findings would hold significant implications for public health which will provide information for the development of policies and strategies aimed at enhancing health monitoring for ICU survivors. Additionally, policymakers can utilize these findings to allocate resources and plan effective care provisions, and delaying the onset of PICS among ICU survivors.

## Methods

### Review registration

The protocol for this systematic review and meta-analysis was registered at PROSPERO (International prospective register of systematic reviews) with registration ID = CRD42024594902.

### Study design and search strategy

This systematic review and meta-analysis was conducted according to the Preferred Reporting Items for Systematic Reviews and Meta-Analyses (PRISMA) 2020 checklist [18]. A systematic search of PubMed/Medline, Embase, CINAHL (Cumulative Index to Nursing and Allied Health Literature), and Scopus data bases was done to find published articles. Additionally, in order to find unpublished studies and grey literature, we searched Google Scholar. All published and gray literature was retrieved, critically evaluated, and assessed to be included in this study spanning to 25 October 2024. The search terms employed using "AND" and "OR" Boolean operators to retrieve articles include: (Post-Intensive Care Syndrome OR PICS) AND (ICU survivors OR Intensive Care Unit survivors) AND (Prevalence OR Epidemiology) AND (Associated factors OR Risk factors OR Determinants OR Contributing factors). The Co, Co, Pop (Condition, Context, and Population) and PEO (Population, Exposure and Outcome) search strategies were used (S1 Table).

### Inclusion criteria and exclusion criteria

This systematic review included all published and unpublished gray literatures reporting PICS and or associated factors published in English spanning to 25 October 2024. Articles without access to the full-text and that did not report on the prevalence of PICS were excluded. At The beginning, each article was evaluated independently for inclusion based on its title and abstract. Then the full-text was assessed to further screen research that included based on the title and abstract review. Duplicated articles were managed by keeping the one with full-text access and removing those without the full-text.

 

## Outcome of interest

The primary outcome was the prevalence of PICS, which is represented as percentage and frequency in articles. It can be one or more of the different domains namely Physical domain: Includes muscle weakness, functional impairments, and reduced exercise capacity, Cognitive domain: Covers impairments in memory, attention, and executive functioning and Mental health domain: Encompasses symptoms of anxiety, depression, and post-traumatic stress disorder (PTSD). There is variability in measurement methods used across studies, particularly in the mental health domain. We have used the reported outcome irrespective of assessment tools used. Based on the PEO (P=Population, E=exposure, O=outcome) model, the secondary outcome was association of PICS with ICU length of stay, which were estimated and provided with odds ratios. The variables used in this meta-analysis to estimate the secondary outcome were those that were declared statistically significant in the included articles.

## Quality assessment and data extraction

The quality of included articles was assessed by the Newcastle Ottawa Scale (NOS) for cohort studies [19]. The tool comprises three categories: Selection with four items (representativeness of the exposed cohort, selection of the non-exposed cohort, ascertainment of exposure to implants, and demonstration that outcome of interest was not present at start of study), comparability with one item (comparability of cohorts on the basis of the design or analysis), and outcome with three items (assessment of outcome, was follow up long enough for outcomes to occur, adequacy of follow up of cohorts).(S2 Table)

Using Microsoft Excel data extraction checklist, the two authors (T.A. and M.A.M.) independently evaluated and extracted the articles for inclusion in the review and overall research quality. Primary author, study year, study region, study design, sample size, prevalence of PICS, post ICU follow-up time, and odds ratio of factors affecting PICS were all included in the data extraction format. Any disagreement amongst the reviewers was settled by dialogue and the participation of other reviewers (G.A., A.G., and M.G.).

## Data analysis

Extracted data using Microsoft Excel was imported to STATA version 17.0 for statistical analysis. The I2 test statistic was used to evaluate the heterogeneity among the studies [20]. Since there is high heterogeneity, we estimated the pooled prevalence of PICS using a random-effects model. We used funnel plot [21] and Egger's [22] test to check for publication bias subjectively and objectively. Furthermore, sources of heterogeneity were assessed using meta-regression of PICS prevalence with publication year, sample size, and post-intensive care follow-up time as covariates, leave-one-out sensitivity analysis, and subgroup analysis by region, study period before COVID-19 vs after COVID-19, study design and admission with COVID-19 vs without COVID-19. The different significant factors associated with PICS in the primary studies were presented using description.

# Results

## Article selection

Initially, a total of 536 published articles was retrieved from different data bases and sources. Then, 267 duplicated records were removed before title and abstract review. After review of the title and abstract, 238 articles were further removed. Further eight articles [23–30] were excluded because the primary outcome is not reported as main outcome or couldn't be estimated from the results of these articles. Moreover, four articles were excluded because of low quality score on NOS [30–33]. In the final analysis, 19 articles were included. (Fig 1).

## Characteristics of included studies

This systematic review included 19 articles [2,14,16,17,34–48]with a total sample size of 10179 participants. Four out of 19 articles were conducted using retrospective cohort design and the rest used prospective cohort study design. These

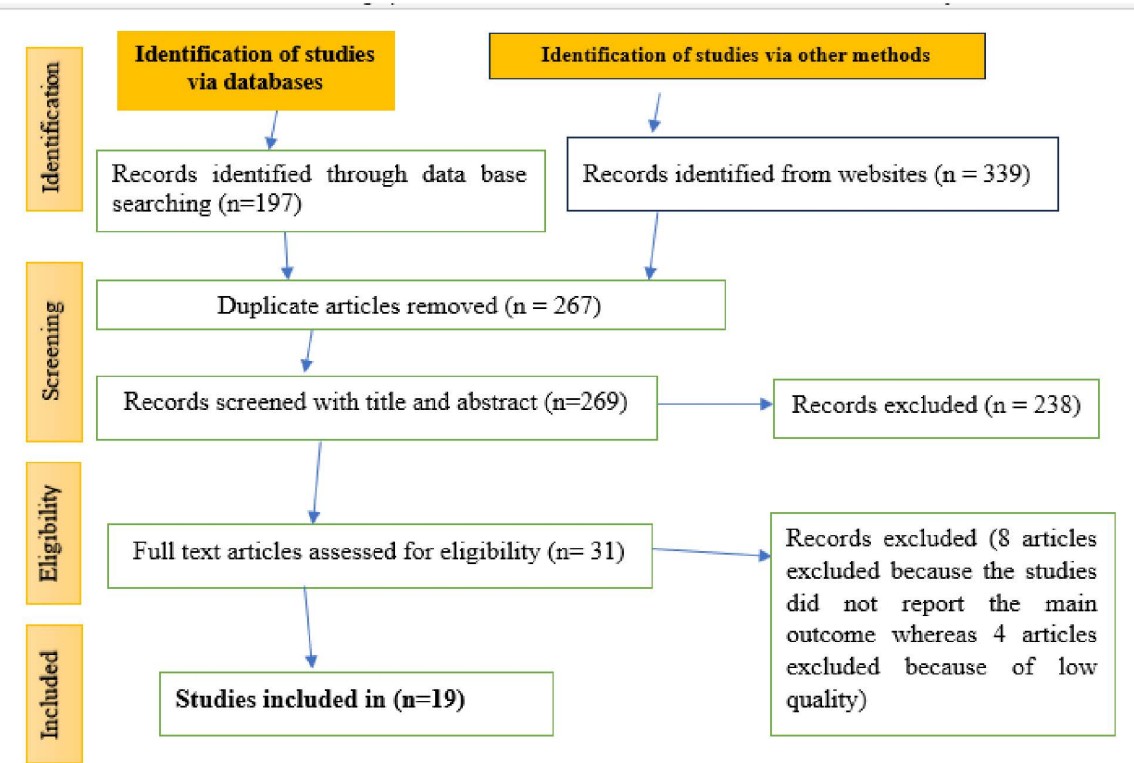

**Fig 1. PRISMA 2020 flow diagram showing article selection process for systematic review of PICS.**

studies were conducted in four regions, namely four in America (south and north), eight in Europe, five in Asia and two in Africa. Regarding study period, 16 out of 19 studies were conducted after the occurrence of COVID-19 pandemic. Nine of the total 19 articles were conducted among patients admitted with COVID-19 diagnosis who survived ICU treatment while the rest 10 were conducted among patients admitted with a diagnosis other than COVID-19 (Table 1).

The quality assessment showed that 13 studies achieved the highest score of 8, while six studies scored 7, one study scored 6. The most common limitations were in representativeness of the exposed cohort and ascertainment of exposure, with some studies, such as Amacher et al. (2024) and Kawakami et al. (2021), missing points in these areas, potentially affecting selection bias and exposure reliability. Additionally, adequacy of follow-up was a concern in a few studies, particularly Abbas et al. (2019) and Alejandro et al. (2024), which could impact outcome completeness. Despite these variations, the majority of studies demonstrated robust designs and appropriate control for confounders, ensuring the overall reliability of the review's conclusions.

Pooled prevalence of post-intensive care syndrome among ICU-survivors.

In this systematic review and meta-analysis, the pooled prevalence of PICS was found to be 54.35 (95% CI = 45.54, 63.15). Using random effects model, high heterogeneity was observed between studies with an I2 test result of 98.47%. As a result, subgroup analysis was performed to identify the source of heterogeneity.

Significant variations in prevalence rates were observed in Alejandro et al. (2024), Friberg et al. (2023), and Weidman et al. (2022) compared to other studies. These differences can largely be attributed to variations in patient populations, with Alejandro et al. (2024) and Weidman et al. (2022) focusing on hospitalized COVID-19 patients, and Friberg et al. (2023) primarily examining non-COVID cases. Additionally, differences in treatment methods and study design—such as

**Table 1. Characteristics of included articles for prevalence of PICS among ICU-survivors.**

| Authors name | Publica-tion Year | Study period | Region | Study design | Follow-up time in months | Type of cases | Sample size | Preva-lence (%) |
|---|---|---|---|---|---|---|---|---|
| Abbas et. al., [16] | 2019 | before COVID | Africa | prospective cohort | 0 | non-COVID | 420 | 52.4 |
| Agha et. al., [45] | 2019 | before COVID | Africa | prospective cohort | 1 | non-COVID | 40 | 50 |
| Alejandro et. al., [37] | 2024 | after COVID | America | Retrospective cohort | 0 | COVID | 126 | 14.3 |
| Amacher et. al., [47] | 2024 | after COVID | Europe | prospective cohort | 24 | non-COVID | 107 | 43 |
| Banno et. al., [35] | 2021 | after COVID | Europe | prospective cohort | 12 | COVID | 18 | 67 |
| Friberg et. al., [44] | 2023 | after COVID | Europe | prospective cohort | 3 | non-COVID | 273 | 19.8 |
| Hatakeyama et. al., [41] | 2022 | after COVID | Asia | prospective cohort | 5.5 | COVID | 251 | 58.6 |
| Hatch et. al., [46] | 2018 | before COVID | Europe | prospective cohort | 3 | non-COVID | 4943 | 55.2 |
| Inoue et.al., [14] | 2022 | after COVID | Asia | prospective cohort | 3 | non-COVID | 77 | 70 |
| Kang et. al., [40] | 2024 | after COVID | Asia | prospective cohort | 3 | non-COVID | 475 | 49.7 |
| Kawakami et. al., [2] | 2021 | after COVID | Asia | prospective cohort | 6 | non-COVID | 96 | 63.5 |
| Maley et. al, [39] | 2022 | after COVID | America | prospective cohort | 6 | COVID | 63 | 80 |
| Martínez et. al., [38] | 2023 | after COVID | America | prospective cohort | 1 | COVID | 22 | 64 |
| Nanwani-Nanwan et. al., [42] | 2022 | after COVID | Europe | prospective cohort | 3 | COVID | 186 | 75 |
| Pun et. al., [36] | 2021 | after COVID | Europe | Retrospective cohort | 1 | COVID | 2088 | 54.9 |
| Rousseau et. al., [34] | 2021 | after COVID | Europe | prospective cohort | 3 | COVID | 42 | 40.6 |
| Tejero-Aranguren et. al., [17] | 2022 | after COVID | Europe | prospective cohort | 3 | non-COVID | 87 | 56.3 |
| Unoki et. al., [48] | 2021 | after COVID | Asia | Retrospective cohort | 12 | non-COVID | 778 | 33.8 |
| Weidman et. al., [43] | 2022 | after COVID | America | Retrospective cohort | 2 | COVID | 87 | 90 |

prospective versus retrospective cohorts—also contributed to the discrepancies in prevalence rates. These factors help explain the extreme variations observed in the reported findings (Fig 2).

In sub-group analysis by region, the highest prevalence was observed in studies done in south and north America with overall prevalence of 61.97 (95% CI = 28.33, 95.62) and I2 of 98.4%. However, in studies done in Africa, the overall prevalence was 52.19 (95% CI = 47.63, 56.76) with lowest heterogeneity of I2 = 0.00. In subgroup analysis by study period, although no difference was observed in overall prevalence between the groups, studies done before COVID-19 pandemic showed minimal heterogeneity with overall prevalence of 54.63 (95% CI = 52.50, 56.75) and I2 of 13.94%. On the other hand, no significant variations were observed in prevalence and heterogeneity in subgroup analysis based on study design and type of patients (COVID-19 vs non-COVID-19). (Table 2).

## Meta regression

Meta regression of prevalence of PICS with sample size post-ICU time in months, and publication year as co-variates was done. However, none of them was found to affect the overall prevalence and therefore, they are not found to be sources of heterogeneity. (Table 3).

   

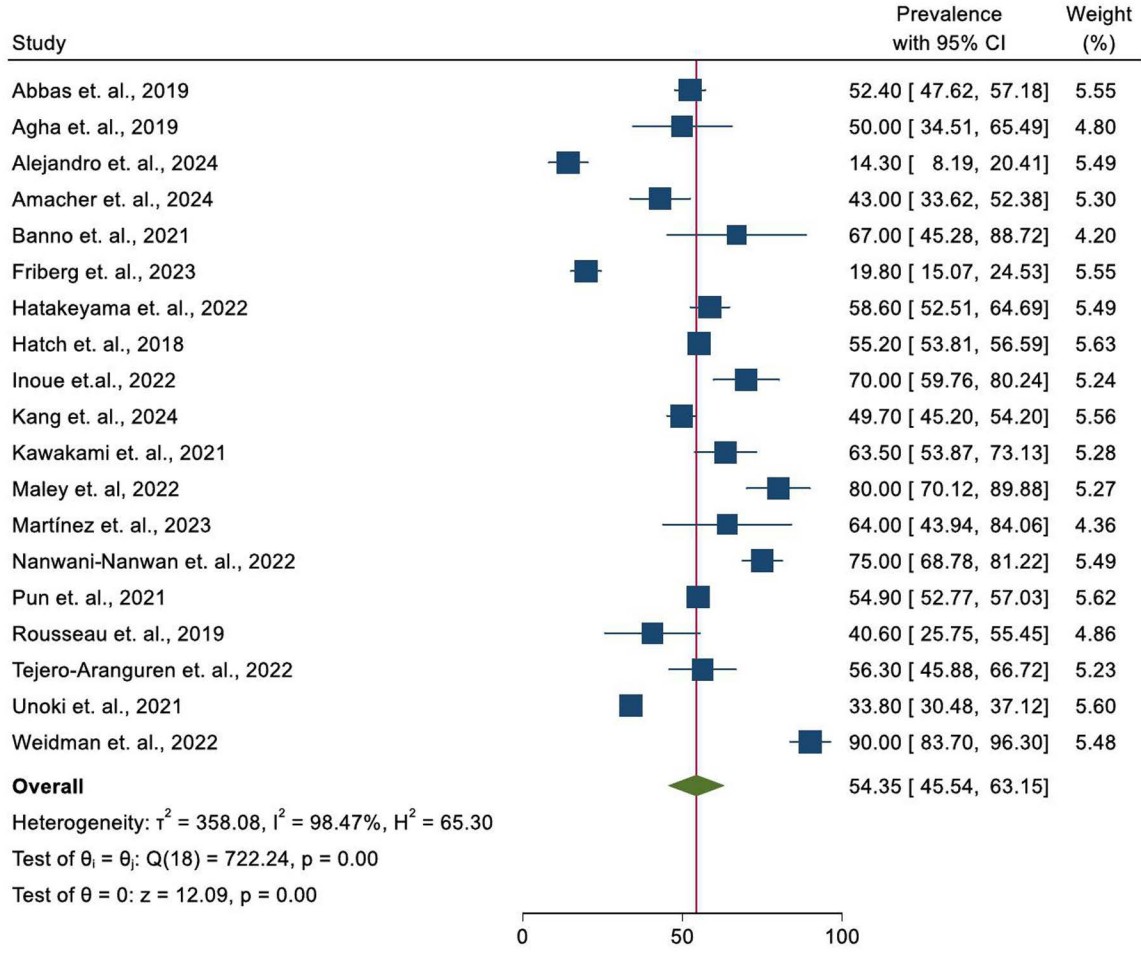

**Fig 2. Pooled prevalence of PICS among ICU-survivors.** The red vertical line represents the overall effect size while the green diamond represents the pooled prevalence. The dark blue squares with horizontal lines represent each study's prevalence and the respected confidence intervals.

### Prevalence among the different domains of PICS among ICU-survivors

Among the three domains of PICS, the highest score was observed in the physical domain with overall prevalence of 45.99 (95% CI = 34.66, 57.31). The cognitive domains of PICS show an overall prevalence of 32.12 (95% CI = 23.68, 40.55) and the mental domain resulted an overall prevalence of 32.36 (95% CI = 25.88, 38.83). (Table 4).

### Leave-one-out sensitivity analysis

The leave one out sensitivity analysis was done to check whether individual studies affected the overall prevalence or not. However, no individual studies had affected the overall effect size. (Fig 3).

### The prevalence of PICS and its association with ICU length of stay among ICU-survivors

Three studies [16,40,41] a cumulative number of patients accounting 162, among a total of 19 reported the length of ICU as a potential determinant factor for the occurrence of PICS among ICU-survivors. In this meta-analysis, those

**Table 2. Subgroup analysis results of prevalence of PICS among ICU-survivors.**

| Sub-groups | Number of studies | Prevalence with 95% CI | I2 | P-value |
|---|---|---|---|---|
| By region | | | | |
| Africa | 2 | 52.19 (47.63, 56.76) | 0.00% | 0.77 |
| America | 4 | 61.97 (28.33, 95.62) | 98.40% | 0.00 |
| Asia | 5 | 54.57 (42.15, 67.00) | 96.76% | 0.00 |
| Europe | 8 | 51.08 (38.95, 63.21) | 96.58 | 0.00 |
| By study period | | | | |
| Before COVID-19 | 3 | 54.63 (52.50,56.75) | 13.94% | 0.00 |
| After COVID-19 | 16 | 54.72 (44.24, 65.20) | 98.05% | 0.00 |
| By type of cases | | | | |
| COVID-19 | 9 | 60.40 (45.10, 75.70) | 97.96% | 0.00 |
| Non-COVID-19 | 10 | 49.00 (39.86, 58.14) | 97.38% | 0.00 |
| By study design | | | | |
| Retrospective cohort | 4 | 48.23 (16.59, 79.88) | 99.63% | 0.00 |
| Prospective cohort | 15 | 55.98 (48.01, 63.94) | 96.13% | 0.00 |

CI = Confidence Interval

**Table 3. Meta regression results of prevalence of PICS with sample size post-ICU time in months, publication year.**

| Variables | Coefficient | Standard error | P-value | 95% CI |
|---|---|---|---|---|
| Sample size | -0.00295 | 0.004733 | 0.53 | (-0.01, 0.01) |
| Post-ICU time in months | -0.08363 | 0.882617 | 0.93 | (-1.81, 1.64) |
| Publication year | -2.87735 | 3.273243 | 0.38 | (-9.29, 3.54) |

**Table 4. Pooled prevalence of different domains of PICS among ICU-survivors.**

| Domains | Number of studies | Prevalence with 95% CI | I2 | P-value |
|---|---|---|---|---|
| Physical domain | 13 [14,16,17,35,38–43,45,47] | 45.99 (34.66, 57.31) | 96.62 | 0.00 |
| Cognitive domain | 14 [2,14,16,17,35,36,38–43,45,47] | 32.12 (23.68, 40.55) | 96.76 | 0.00 |
| Mental domain | 16 [2,14,16,17,35,38–48] | 32.36 (25.88, 38.83) | 95.87 | 0.00 |

CI = Confidence Interval.

patients who stayed more than four days in the intensive care unit were 1.207 [OR = 1.207; 95% CI = 1.119, 1.295] times more likely to develop at least one among the three domains of PICS in the post-intensive care period than their counterparts. (Fig 4).

## Publication bias

Publication bias was assessed using standard funnel plot symmetry test and Egger's test. Although the Egger's test didn't indicate the presence of publication bias (p = 0.3680), the funnel plot showed asymmetrical appearance indicating the presence of subtle publication bias. Therefore, trim-fill analysis was performed. The trim-fill analysis adjusted the symmetry of the funnel plot by inputting of five studies to the left. The overall prevalence was found to be 46.55 (95% CI = 37.06, 56.04) if five studies were inputted, indicating that there was likelihood of publication of overreported studies. (Figs 5–6).

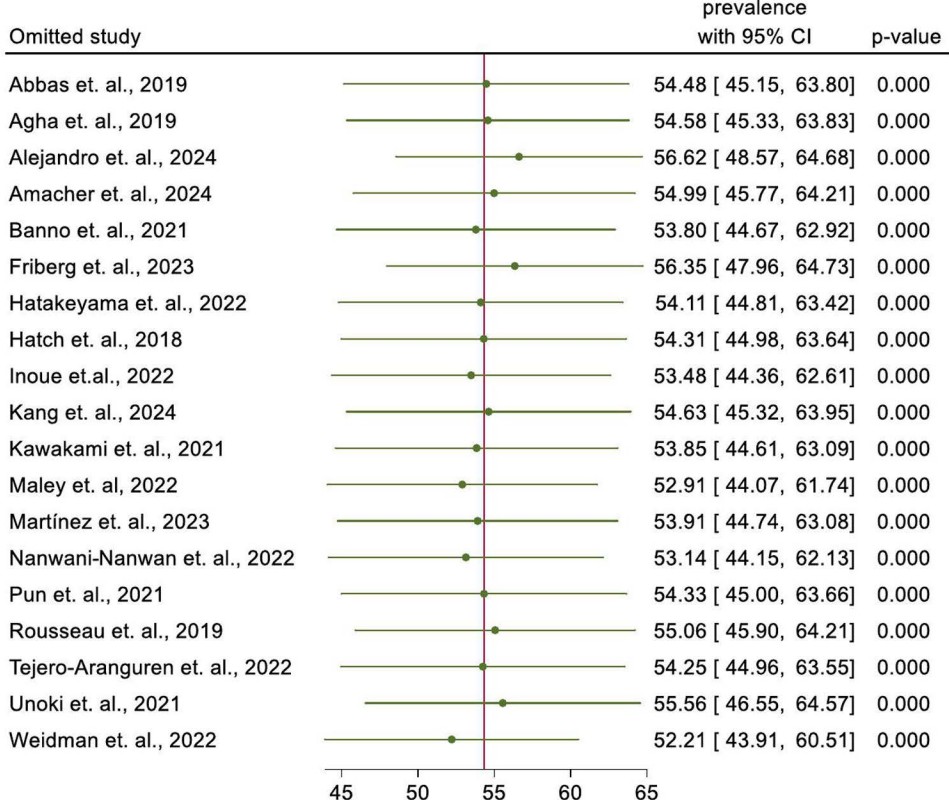

**Fig 3. Leave-one-out sensitivity analysis of the pooled prevalence of PICS among ICU-survivors.** It tells what the prevalence would be when each of the studies in the figure were removed.

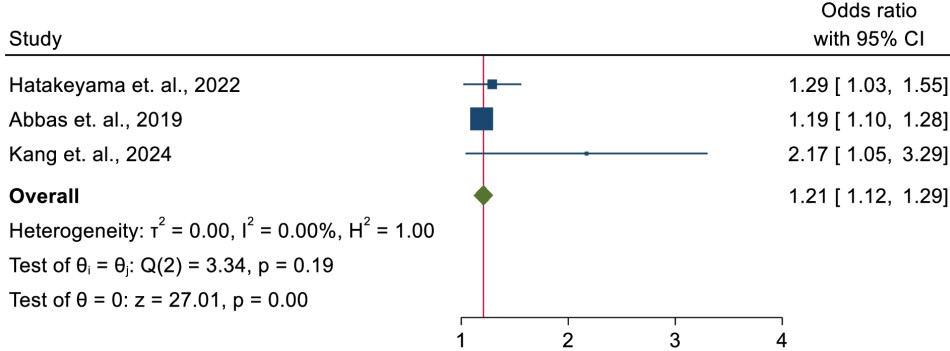

**Fig 4. Shows the association between PICS and ICU length of stay.** The red vertical line represents the overall effect size while the green diamond represents the pooled odds ratio. The dark blue squares with horizontal lines represent each study's odds ratios and the respected confidence intervals.

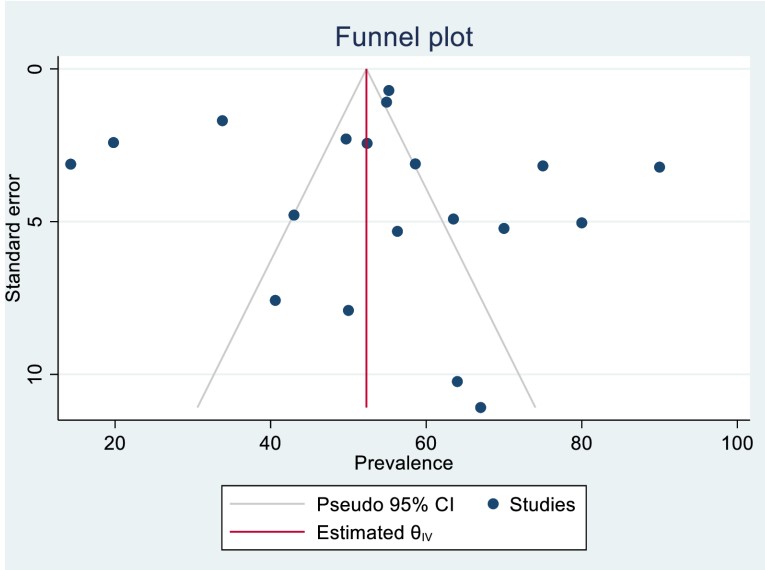

**Fig 5. Funnel plot for assessing the presence of publication bias in the prevalence of PICS among ICU-survivors.** Asymmetric distribution of the studies against the estimated effect line shows publication presence of bias.

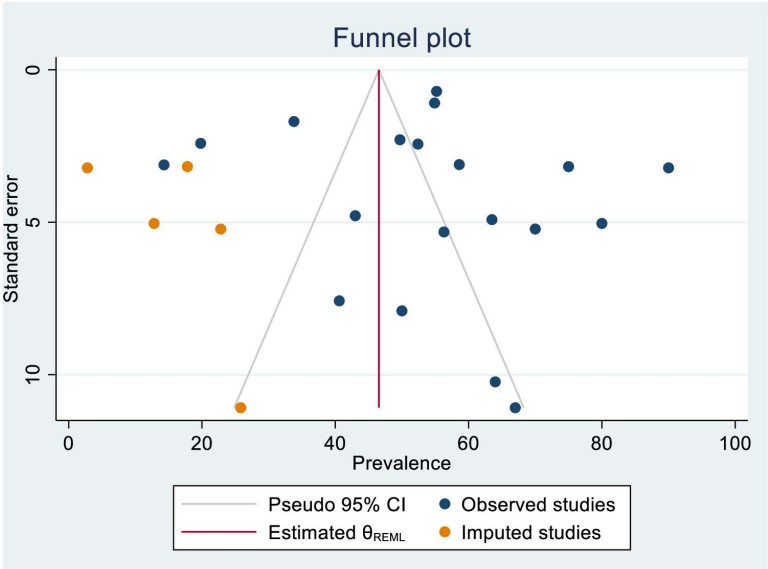

**Fig 6. Trim and fill analysis of publication bias in the prevalence of PICS among ICU-survivors.** The hypothetical addition of studies indicated with five orange dots made the plot symmetric meaning that five studies are missed.

## Discussion

Most ICU survivors experience mild and long-term impairment that impact their quality of life. Therefore, this systematic review and meta-analysis study is aimed to generate comprehensive evidence about the global prevalence of post-intensive care syndrome and its association with ICU length of stay among ICU-survivors.

This systematic review and meta-analysis found that the pooled global prevalence of PICS was 54.35% (95% CI = 45.54, 63.15), which indicates that one out of two critically-ill patients may experience PICS. A recent systematic review supported this finding, which reported that about 50–80% of the ICU survivors had mild or long-term impairments [49]. Moreover, this finding aligns with previous research that has reported that the prevalence rates for PICS, often between 25% and 70%, depending on the population studied and methodological differences [5,6]. The high prevalence observed in this study underscores the significant burden that PICS places on ICU survivors and highlights the need for further research and interventions aimed at improving post-ICU care. Thus, multidisciplinary rehabilitation of ICU-survivors is recommended, which aimed to reduce the long-term complications [50,51].

One of the most notable results of this analysis was the high heterogeneity (I² = 98.47%) observed across studies, suggesting substantial variability in prevalence estimates. This could be attributed to differences in geographical regions, patient characteristics, study designs, and methods used to diagnose PICS. To explore the sources of this heterogeneity, subgroup analyses were performed based on region, study period, patient type (COVID-19 vs. non-COVID-19), and study design.

## Regional differences in PICS prevalence

In sub-group analysis by region, the highest prevalence was observed in studies done in South and North America with overall prevalence of 61.95%, with high heterogeneity (I² = 98.4%), followed by Asia, Africa, and Europe 54.57%, 52.19%, and 51.08%, respectively. This difference can be attributed to the difference in enrolling criteria and PICS definitions, and probably due to the methodological heterogeneity between studies [52]. Moreover, the prevalence of PICS can be varied depending on the time after discharge and the characteristics of study participants [53]. Thus, the higher prevalence in the Americas may be due to the time of post-discharge follow-up, as has been noted in other studies comparing global ICU outcomes [54–56]. The prevalence in African studies was lower, with no heterogeneity (I² = 0.00%). The low heterogeneity observed in African studies could be attributed to the limited number of studies (n = 2) and potentially more homogenous healthcare settings compared to the more diverse healthcare systems and patient characteristics in the Americas.

## The effect of study period and COVID-19 on prevalence of PICS

The impact of the COVID-19 pandemic on PICS prevalence is another critical finding. Although no significant difference in prevalence was observed between studies conducted before and after the pandemic, heterogeneity was markedly lower in studies conducted before COVID-19 (I² = 13.94%). This suggests that the pandemic may have introduced additional variability in the risk of PICS due to factors such as increased ICU admissions, changes in care practices, and the direct effects of COVID-19 on physical, cognitive, and psychological health. Previous studies have indicated that COVID-19 survivors, especially those requiring ICU care, are at a higher risk of developing PICS, given the severity of illness and the prolonged ICU stays often required [57].

## PICS domains

Among the three domains of PICS, the highest score was observed in the physical domain with overall prevalence of 45.99%, consistent with previous studies that have shown significant long-term physical impairments in ICU survivors, such as muscle weakness and fatigue [58]. A recent study reported similar finding in which physical impairment is the most common and can be severe, significantly affecting the quality of life of critically-ill patients [59]. Moreover, this is also in line with the findings that revealed physical impairment is present in 25–80% of adult ICU survivors [60], and many ICU survivors have persistent physical impairment, including muscle weakness, neuropathy, and/or myopathy, and muscle disuse atrophy [61,62]. It might be due to the risk factors for critical illness- related neuromuscular abnormalities including prolonged length of stay, sepsis, multi-organ dysfunctions, renal replacement therapy,

and administration of vasopressors [63]. Moreover, physical impairment, mostly resulting from muscle weakness, is a well-known long term sequalae among ICU-survivors, which leads trouble in returning to work after hospital discharge [64,65]. This suggest the importance of early mobilization and physical rehabilitation, which is supported by an evidence that survivors who received rehabilitation after discharge from the hospital had lower mortality risk than who did not [51,66]. Therefore, increased understanding of risk factors for different domains of PICS, especially physical impairment has to be improved to boost the healthcare providers' ability to identify potentially high-risk patients for screening and intervention.

The cognitive and mental health domains had slightly lower prevalence rates (32.12% and 32.36%, respectively), but these domains remain critical aspects of PICS, as survivors often experience cognitive decline and psychological issues such as anxiety, depression, and Post Traumatic Stress Disorder (PTSD) [5]. These findings highlight the need for a multidisciplinary approach to post-ICU care, addressing not only physical recovery but also cognitive and mental health rehabilitation.

### ICU length of stay and PICS

Although the analysis was limited by the number of included articles, it was found that those patients who stayed more than four days in the intensive care unit were 1.2 times more likely to develop PICS in at least one domain of in the post-intensive care period than their counterparts. This finding is consistent with previous studies suggesting that prolonged ICU stays, due to factors such as mechanical ventilation, sedation, and immobilization, contribute to the risk of long-term physical and psychological impairments [60,67]. Longer ICU stays are often associated with more severe illnesses and complications such as delirium and critical illness polyneuropathy, both of which are known to increase the risk of PICS [12].

### Limitations and publication bias

Several limitations were encountered during this analysis that warrant consideration. First, we found a high degree of inconsistencies between instruments used to assess different domains of PICS. This inconsistency may affect the comparability of findings across studies. Second, although the Egger's test did not show evidence of publication bias, the asymmetry of the funnel plot suggests that subtle publication bias may be present. The trim-and-fill analysis adjusted for this potential bias, resulting in a slightly lower prevalence estimate of 46.55%. This indicates that the true prevalence of PICS may be slightly overestimated in published studies, possibly due to the underreporting of studies with lower prevalence rates. Despite this, the overall findings of the meta-analysis remain robust, as no individual study was found to significantly influence the pooled prevalence in the leave-one-out sensitivity analysis. Furthermore, this study represents the first attempt to summarize the global prevalence of PICS, highlighting the importance of further research to validate and refine these estimates.

### Conclusion

This systematic review and meta-analysis demonstrate a high prevalence of PICS among ICU survivors, particularly in regions such as the Americas, and highlight the possible association between ICU length of stay and the development of PICS. These findings underscore the need for targeted interventions to mitigate the long-term effects of critical illness, particularly for patients with prolonged ICU stays. Future research should focus on refining post-ICU care protocols to address the physical, cognitive, and psychological domains of PICS. Additionally, more studies are needed to evaluate the relationship between ICU length of stay and PICS development, considering factors such as age, gender, and length of post-ICU hospital stay. Expanding research in these areas will provide a more comprehensive understanding of risk factors and help guide targeted interventions for PICS prevention and management.

## Supporting information

**S1 File. Supplementary information.rar: It contains S1 Table, S2 Table, Data Set, Numbered list of literature search result, and the PRISMA_2020 checklist.**
(DOCX)

## Acknowledgments

Our gratitude goes to all individual at Debre Markos University, College of Health Sciences and School of Medicine, who assisted us in this review.

## Author contributions

**Conceptualization:** Temesgen Ayenew, Menberu Gete, Haile Amha, Mengistu Abebe Messelu.

**Data curation:** Temesgen Ayenew, Menberu Gete, Mengistu Abebe Messelu.

**Formal analysis:** Temesgen Ayenew, Mengistu Abebe Messelu.

**Methodology:** Temesgen Ayenew, Menberu Gete, Haile Amha.

**Software:** Temesgen Ayenew, Mengistu Abebe Messelu.

**Supervision:** Abebe Dilie Afenigus, Afework Edmealem, Girma Alem, Bekele Getenet Tiruneh.

**Validation:** Temesgen Ayenew, Mihretie Gedfew, Addisu Getie, Abebe Dilie Afenigus, Afework Edmealem, Haile Amha, Girma Alem, Bekele Getenet Tiruneh, Mengistu Abebe Messelu.

**Visualization:** Temesgen Ayenew, Mihretie Gedfew, Addisu Getie, Abebe Dilie Afenigus, Afework Edmealem, Haile Amha, Girma Alem, Bekele Getenet Tiruneh, Mengistu Abebe Messelu.

**Writing – original draft:** Temesgen Ayenew, Menberu Gete, Mengistu Abebe Messelu.

**Writing – review & editing:** Mihretie Gedfew, Addisu Getie, Abebe Dilie Afenigus, Afework Edmealem, Haile Amha, Girma Alem, Bekele Getenet Tiruneh.

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
