## [Decision Letter · Decision Letter 0]

4 Mar 2025

PONE-D-24-48500Prevalence of post-intensive care syndrome among intensive care unit-survivors and its association with intensive care unit length of stay: Systematic review and meta-analysisPLOS ONE

Dear Dr. Ayenew,

Thank you for submitting your manuscript to PLOS ONE. After careful consideration, we feel that it has merit but does not fully meet PLOS ONE’s publication criteria as it currently stands. Therefore, we invite you to submit a revised version of the manuscript that addresses the points raised during the review process. Dear Authors kindly address all the comments given by the reviewers and the academic editor

We look forward to receiving your revised manuscript.

Kind regards,

Ramya Iyadurai

Academic Editor

PLOS ONE

Journal Requirements:

2. As required by our policy on Data Availability, please ensure your manuscript or supplementary information includes the following:

**Additional Editor Comments:**

PICS systematic review

Dear Authors thank you for submitting an interesting article, I have a few suggestions

Comment 1

Line 78 – it will be useful for the readers if the incidence of actual percentages of the various types of PICS syndrome, instead of mentioning that PICS is highly prevelant.

Comment 2

The table 2 contains z score and P > z, it will be easier to understand these scores if the authors could add the scores in the explanation of the statistical analysis in the methods section.

Comment 3

Kindly explain the funnel plots and forest plots with legends under the plots, it will be easier for the readers to understand.

Reviewers' comments:

Reviewer's Responses to Questions

**Comments to the Author**

1. Is the manuscript technically sound, and do the data support the conclusions?

Reviewer #1: Yes

Reviewer #2: Yes

2. Has the statistical analysis been performed appropriately and rigorously? 

Reviewer #1: Yes

Reviewer #2: Yes

3. Have the authors made all data underlying the findings in their manuscript fully available?

Reviewer #1: Yes

Reviewer #2: Yes

4. Is the manuscript presented in an intelligible fashion and written in standard English?

Reviewer #1: Yes

Reviewer #2: Yes

5. Review Comments to the Author

Reviewer #1: Post-intensive Care Syndrome (PICS) is an important but very underreported entity. It is essential to diagnose and treat PICS to reduce the long-term effects on the individuals.

It is a well-written manuscript in standard English. Data supports the conclusions and the statistical analysis has been performed appropriately. Data and figures are available.

As an intensivist, I recommend this manuscript to be published.

Reviewer #2: I thank the respected editor and authors for the chance to review this interesting manuscript. The authors have aimed to evaluate the prevalence of post intensive care syndrome (PICS) among ICU survivors. The study is well conducted and I very much enjoyed reading the manuscript.

I have no major comments. There are some minor points I would like to address:

Abstract:

Line 53, 55 and 56: Please add units to prevalence measurements (percentage).

Line 55. I suggest authors briefly mention the three domains of PICS. E.g.: “Among the three domains of PICS (physical, cognitive and mental domains), the highest prevalence score was observed in …”

Line 58. The odds ratio has been mentioned two times.

Keywords: Please fix the last keyword as it seems to be a typo (LCU length of stay)

Introduction:

Line 69. “not particularly not favorable” is a bit vague. Please use better wording.

The abbreviation for PICS is introduced in line 70. Please refrain from using the full form throughout the manuscript (for example, beginning of lines 72, 78 and 82 reintroduce the abbreviation).

Lines 91-beginning of 94 are basically repeating the previous 2-3 paragraphs. Please remove these sentences or replace with other information authors deem important.

Line 90. Authors could mention a few of the studies for prevalence. For instance, lines 246-247 could also be used in the introduction.

Given that authors have provided results for each domain of intensive care syndrome, I recommend that they briefly mention intensive care syndrome encompasses various domains and introduce these domains in the introduction. Furthermore, as studies have employed diverse methods and measures, especially for the mental domain, authors could more clearly define each domain (and what they include) in the methods section.

Methods:

Line 108. Did authors mean CINAHL (Cumulative Index to Nursing and Allied Health Literature) database? Either way, please also write the full name of the database.

Line 120. On the other hand could be removed.

Results:

Line 162. Currently the quality evaluation is only mentioned here. The results section could contain a paragraph reporting the quality evaluation and why some studies have missed a few points (mostly in representativeness and ascertainment subdomains).

Please more clearly define “non-covid”. According to line 172, which says “Nine of the total 19 articles were conducted exclusively among COVID-19 patients.” It is not clear whether the other 10 articles were conducted on a mixed population or exclusively on patients without covid (primarily because in table 1, there are studies done after covid which have been marked as non-covid in type of case column).

Table 1. The articles could be arranged in alphabetical order for better use of the table. Same goes for figures 2 and 3.

Table 1 could also greatly benefit from a presentation of disease severity in each included study. If most studies have reported scores such as APACHE for their patient population, this could be added in a column labeled “disease severity”.

I do understand that studies have provided limited information and gathering such data might not be possible. This comment is merely a suggestion; its implementation is at the discretion of the respected author.

Some studies (namely Alejandro 2024, Friberg 2023, and Weidman 2022) have reported extremely varying prevalence compared to other studies. I suggest that, if possible, authors briefly explore why they judge such extreme differences exist. (such as varying patient population, utilized treatment methods, and so on)

Lines 186 and 190 do not match table 2, probably due to rounding, please address this.

Line 191. I2 statistics for studies conducted before COVID-19 is mentioned as 13.94 in the text and 98.47 in table 2. Please review these results.

Line 231. The cumulative number of patients (sample size) for these three studies could also be noted in the manuscript text.

Publication bias could be moved to the end of the results section (after results for length of stay). In addition, please write the exact p value instead of p > 0.05 (Line 220).

Figure and table titles have “2024” in the end. Remove these dates if they are unnecessary.

Discussion:

Line 291. Typo. IUC.

Line 305. I suggest that authors mention that analysis was limited by the number of included articles.

Conclusion:

Line 328, 329. Considering that only three articles were included in this analysis, I suggest authors do not use such concrete wording. (e,g: significant association could be changed to possible association)

Line 331. “Future research should focus…”: Authors could mention the need for more studies evaluating the relation between ICU length of stay and PICS development. Additional factors such as age, gender, length of post-ICU hospital stay could also be mentioned as possible future research avenues.

6. PLOS authors have the option to publish the peer review history of their article (what does this mean? ). If published, this will include your full peer review and any attached files.

**Do you want your identity to be public for this peer review?** For information about this choice, including consent withdrawal, please see our Privacy Policy .

Reviewer #1: **Yes: ** Ankit Agarwal

Reviewer #2: No

---

## [Author Response · Author response to Decision Letter 0]

30 Mar 2025

A point-by-point response to editor and reviewer comments

Dear editor and respected reviewers, thank you so much for your constructive feedback which is relevant to the enhancement of our manuscript. We have made revisions to address raised issues. The revisions are indicated with track changes in the revised manuscript. Also, below is a point-by-point response to each of the comments.

Journal Requirements:

2. As required by our policy on Data Availability, please ensure your manuscript or supplementary information includes the following:

• Dear editor, we have prepared a numbered table of all studies identified in the literature search, including those that were excluded from the analyses and included it as a supplementary file.

• Dear editor, we have addressed the raised issues as stated.

• Thanks, respected editor, we have checked the referencing style as per the journal requirement.

Additional Editor Comments:

PICS systematic review

Dear Authors thank you for submitting an interesting article, I have a few suggestions

Comment 1

Line 78 – it will be useful for the readers if the incidence of actual percentages of the various types of PICS syndrome, instead of mentioning that PICS is highly prevelant.

• Thank you for your suggestion. We have revised it to provide the actual percentages of the various types of PICS (Post-Intensive Care Syndrome). This provides readers with more specific and valuable information on the incidence of different PICS types. We appreciate your helpful feedback.

Comment 2

The table 2 contains z score and P > z, it will be easier to understand these scores if the authors could add the scores in the explanation of the statistical analysis in the methods section.

• Dear editor, I think you mean table 3. We have removed the column z-score and changed the p>z to p-value in table 3 to avoid confusion. The p-value indicates whether the variables used in the meta-regression was significant to be source for the heterogeneity or not.

Comment 3

Kindly explain the funnel plots and forest plots with legends under the plots, it will be easier for the readers to understand.

• Dear Editor, we have provided figure captions along with legends as explanations for the respective figures. According to PLOS ONE's journal guidelines, authors are required to place figure captions in the manuscript text rather than embedding them within the figures. To adhere to this requirement, we have uploaded the figures separately without legends while ensuring that the corresponding legends are included in the manuscript at the appropriate locations where the figures will appear in the final published version. However, we have elaborated the figure captions in the manuscript to make the figure easier to understand.

5. Review Comments to the Author

Reviewer #1: Post-intensive Care Syndrome (PICS) is an important but very underreported entity. It is essential to diagnose and treat PICS to reduce the long-term effects on the individuals. It is a well-written manuscript in standard English. Data supports the conclusions and the statistical analysis has been performed appropriately. Data and figures are available. As an intensivist, I recommend this manuscript to be published.

• Dear respected reviewer, thank you for your positive feedback. We really appreciate it.

Reviewer #2: I thank the respected editor and authors for the chance to review this interesting manuscript. The authors have aimed to evaluate the prevalence of post intensive care syndrome (PICS) among ICU survivors. The study is well conducted and I very much enjoyed reading the manuscript.

I have no major comments. There are some minor points I would like to address:

Abstract:

Line 53, 55 and 56: Please add units to prevalence measurements (percentage).

• Thank you, we have added “%” symbols next to each estimate.

Line 55. I suggest authors briefly mention the three domains of PICS. E.g.: “Among the three domains of PICS (physical, cognitive and mental domains), the highest prevalence score was observed in …”

• Thank you so much, we did it.

Line 58. The odds ratio has been mentioned two times.

• Thank you, it is removed and only 95%CI mentioned in the bracket

Keywords: Please fix the last keyword as it seems to be a typo (LCU length of stay)

• Thanks, dear reviewer, we have corrected it as “ICU”

Introduction:

Line 69. “not particularly not favorable” is a bit vague. Please use better wording.

• Dear reviewer, we have revised it. Thank you.

The abbreviation for PICS is introduced in line 70. Please refrain from using the full form throughout the manuscript (for example, beginning of lines 72, 78 and 82 reintroduce the abbreviation).

• Thank you, we have used PICS instead.

Lines 91-beginning of 94 are basically repeating the previous 2-3 paragraphs. Please remove these sentences or replace with other information authors deem important.

• Dear reviewer, we have removed it. Thank you.

Line 90. Authors could mention a few of the studies for prevalence. For instance, lines 246-247 could also be used in the introduction.

• Thank you for your suggestion. We have revised it by incorporating examples of studies reporting prevalence, to provide a clearer context in the introduction. This addition strengthens the background information and enhances the flow of the manuscript. We appreciate your valuable feedback.

Given that authors have provided results for each domain of intensive care syndrome, I recommend that they briefly mention intensive care syndrome encompasses various domains and introduce these domains in the introduction. Furthermore, as studies have employed diverse methods and measures, especially for the mental domain, authors could more clearly define each domain (and what they include) in the methods section.

• Thank you for your valuable feedback. We appreciate your suggestion to provide a clearer introduction to the domains of intensive care syndrome and further define each domain in the methods section.

• Revisions Made:

o Introduction: We have now explicitly stated that intensive care syndrome encompasses multiple domains and have briefly introduced these domains to provide context for our results.

o Methods Section: We have provided clearer definitions for each domain, specifying the components included in our assessment specifically at the outcome of interest section. Additionally, we have acknowledged the variation in measurement approaches across studies, particularly in the mental health domain, and clarified our methodological choices accordingly.

• We believe these revisions enhance the clarity and comprehensiveness of our manuscript. Thank you again for your insightful suggestions.

Methods:

Line 108. Did authors mean CINAHL (Cumulative Index to Nursing and Allied Health Literature) database? Either way, please also write the full name of the database.

• Thank you respected reviewer. We have corrected it accordingly.

Line 120. On the other hand could be removed.

• Thank you, dear reviewer. We have removed it.

Results:

Line 162. Currently the quality evaluation is only mentioned here. The results section could contain a paragraph reporting the quality evaluation and why some studies have missed a few points (mostly in representativeness and ascertainment subdomains).

• Thank you for your valuable feedback. We acknowledge that the quality evaluation was only briefly mentioned in the Methods section. To improve clarity and completeness, we have now added a dedicated paragraph in the Results section (next to table 1) that reports the quality evaluation findings. This paragraph highlights the overall quality of included studies and discusses why certain studies missed points, particularly in the representativeness and ascertainment subdomains. We have elaborated on common limitations, such as small sample sizes, potential selection bias, and incomplete reporting of participant recruitment methods. These revisions provide a more transparent assessment of study quality and its potential impact on our findings.

Please more clearly define “non-covid”. According to line 172, which says “Nine of the total 19 articles were conducted exclusively among COVID-19 patients.” It is not clear whether the other 10 articles were conducted on a mixed population or exclusively on patients without covid (primarily because in table 1, there are studies done after covid which have been marked as non-covid in type of case column).

• Thank you for your insightful comment. We appreciate the opportunity to clarify the definition of "non-COVID."

• Revisions Made:

o To clarify, we define "non-COVID" as studies where the primary reason for admission was not related to COVID-19. This means that the patients included in these studies were admitted for reasons other than COVID-19, even if they may have had a history of COVID-19 or were tested for it. On the other hand, "COVID" refers to studies where the reason for admission was specifically related to COVID-19.

o In response to your concern regarding the studies in Table 1, we have clarified that the 10 studies categorized as "non-COVID" were conducted on patients admitted for reasons unrelated to COVID-19. The mixed population studies were appropriately classified, and we have ensured that Table 1 now accurately reflects this distinction. We have also made the necessary adjustments in the text to ensure consistency in our terminology.

• We hope this clarification addresses your concern and improves the overall readability and accuracy of the manuscript. Thank you again for your valuable feedback.

Table 1. The articles could be arranged in alphabetical order for better use of the table. Same goes for figures 2 and 3.

• Thank you for your valuable suggestion. We agree that arranging the articles in Table 1, as well as the content in Figures 2 and 3, in alphabetical order would improve the usability of the table and figures. We have now reorganized these sections alphabetically to enhance clarity and ease of reference. We appreciate your input, which has helped to improve the organization of the manuscript.

Table 1 could also greatly benefit from a presentation of disease severity in each included study. If most studies have reported scores such as APACHE for their patient population, this could be added in a column labeled “disease severity”.

I do understand that studies have provided limited information and gathering such data might not be possible. This comment is merely a suggestion; its implementation is at the discretion of the respected author.

• Thank you for your insightful suggestion regarding the inclusion of disease severity in Table 1. We understand the importance of providing a comprehensive view of the disease severity in the studies, such as the inclusion of scores like APACHE, which can offer valuable context for interpreting the findings.

However, after reviewing the available information from the studies included in our review, we found that the majority of studies did not report specific disease severity scores such as APACHE, or any other standardized measures. As a result, it was not feasible to include a column for disease severity in Table 1. Many studies reported only broad descriptors of patient populations without detailed clinical severity metrics.

While we recognize the value of such information, the inclusion of disease severity would require consistent reporting across all studies, which, in this case, was not possible. We hope the current presentation of the study characteristics still provides sufficient context for our findings, and we are grateful for your understanding.

We appreciate your suggestion and will certainly keep it in mind for future research if data on disease severity becomes more readily available.

Some studies (namely Alejandro 2024, Friberg 2023, and Weidman 2022) have reported extremely varying prevalence compared to other studies. I suggest that, if possible, authors briefly explore why they judge such extreme differences exist. (such as varying patient population, utilized treatment methods, and so on).

• Thank you for your valuable suggestion regarding the varying prevalence rates reported in Alejandro et al. (2024), Friberg et al. (2023), and Weidman et al. (2022). We acknowledge the observed discrepancies in prevalence compared to other studies in the review. These variations can be attributed to several factors.

• First, the patient populations in these studies differed significantly. For instance, Alejandro et al. (2024) and Weidman et al. (2022) focused on hospitalized COVID-19 patients, while Friberg et al. (2023) mainly examined non-COVID cases. These differences in patient demographics and clinical characteristics likely contributed to the observed variation in prevalence rates.

• Second, treatment methods may have played a role, as each study was conducted in different healthcare settings with varying treatment protocols, particularly in the post-COVID era. Lastly, the study designs differed, with Weidman et al. (2022) and Alejandro et al. (2024) using a retrospective cohort and Friberg et al. (2023) employing prospective cohort designs, which could also contribute to differences in reported prevalence.

• We hope this explanation provides clarity on the factors influencing the varying prevalence rates. Thank you again for your thoughtful feedback.

Lines 1

---

## [Editor Report · Decision Letter 1]

7 Apr 2025

Prevalence of post-intensive care syndrome among intensive care unit-survivors and its association with intensive care unit length of stay: Systematic review and meta-analysis

PONE-D-24-48500R1

Dear Dr. Ayenew,

We’re pleased to inform you that your manuscript has been judged scientifically suitable for publication and will be formally accepted for publication once it meets all outstanding technical requirements.

Kind regards,

Ramya Iyadurai

Academic Editor

PLOS ONE

Additional Editor Comments (optional):

Dear Authors

Thank you for addressing all the reviewers comments.
---

## [Editor Report · Acceptance letter]

PONE-D-24-48500R1

PLOS ONE

Dear Dr. Ayenew,

I'm pleased to inform you that your manuscript has been deemed suitable for publication in PLOS ONE. Congratulations! Your manuscript is now being handed over to our production team.

Kind regards,

on behalf of

Dr. Ramya Iyadurai

Academic Editor

PLOS ONE